# Experimental Evidence for Partially Dehydrogenated ε-FeOOH

**Yukai Zhuang [1], Zhongxun Cui [1], Dongzhou Zhang [2], Jin Liu [1], Renbiao Tao [3] and Qingyang Hu [1,\*]**

[1] Center for High Pressure Science and Technology Advanced Research, Beijing 100094, China
[2] Hawai'i Institute of Geophysics and Planetology, School of Ocean and Earth Science and Technology, University of Hawai'i at Manoa, Honolulu, HI 96822, USA
[3] Geophysical Laboratory, Carnegie Institution of Washington, Washington, DC 20015, USA
[\*] Correspondence: qingyang.hu@hpstar.ac.cn

**Abstract:** Hydrogen in hydrous minerals becomes highly mobile as it approaches the geotherm of the lower mantle. Its diffusion and transportation behaviors under high pressure are important in order to understand the crystallographic properties of hydrous minerals. However, they are difficult to characterize due to the limit of weak X-ray signals from hydrogen. In this study, we measured the volume changes of hydrous ε-FeOOH under quasi-hydrostatic and non-hydrostatic conditions. Its equation of states was set as the cap line to compare with ε-FeOOH reheated and decompression from the higher pressure pyrite-FeO$_2$H$x$ phase with $0 < x < 1$. We found the volumes of those re-crystallized ε-FeOOH were generally 2.2% to 2.7% lower than fully hydrogenated ε-FeOOH. Our observations indicated that ε-FeOOH transformed from pyrite-FeO$_2$H$x$ may inherit the hydrogen loss that occurred at the pyrite-phase. Hydrous minerals with partial dehydrogenation like ε-FeOOH$x$ may bring it to a shallower depth (e.g., < 1700 km) of the lower mantle.

**Keywords:** ε-FeOOH; high pressure; synchrotron X-ray diffraction; dehydrogenation

## 1. Introduction

The major components of the lower mantle are made of nominally anhydrous minerals, which contain no more than 1 weight percent of water [1,2]. However, recent discoveries of deep hydrous phase (DHP) including δ-AlOOH [3–5], phase H [6], HH-phase [7], and pyrite-type FeO$_2$H$x$ [8,9], which were synthesized at conditions of cold mantle geotherm from natural minerals like diaspore (α-AlOOH) and goethite (α-FeOOH), provide possible mechanisms to transport a significant amount of water to the bottom of the mantle. The potential presence of hydrogen is likely to contribute a variety of seismological features observed at Earth's lower mantle. For example, dehydration melting at the top of the lower mantle could dramatically decrease the seismic velocities below a depth of 660 kilometers [10]. Accumulation of iron-enriched hydrous pyrite-type phase would reduce the speed of seismic waves at the core-mantle boundary, which may be detected at large low shear velocity provinces and ultra-low velocity zones [11–13]. Reservoirs of H also produce hydrides that would possibly infiltrate to the outer core [9,14]. Although H is an influential volatile component, the budget of H in the lower mantle is still under debate [15]. Large uncertainties in the abundance are probably due to the scarcity to find natural samples derived from the deep mantle [16]. However, the number of DHPs revealed by laboratory experiments continues to grow with the development of high-pressure synchrotron-based experiments [17]. The extraordinary thermal stabilities of DHPs suggest that the lower mantle can hold more water than previously expected.

Discoveries of novel DHPs have attracted an appreciable amount of research efforts [5,18–20]. What is equally important is the diffusion and transportation behaviors of hydrogen in those DHPs [21]. Since hydrogen is the lightest element and is highly mobile, an outstanding question is how we can quantify the hydrogen content in DHP under high pressure. It becomes even more challenging when DHP is partially dehydrogenated [22,23]. In this case, we measured the equation of state of $\varepsilon$-FeOOH, a typical DHP in the Fe-O-H ternary system, and studied its hydrogen content based on its volume variation. This method has been testified to determine the hydrogen content in hydride [24] and pyrite-type $FeO_2Hx$ [22]. We provide evidence that the $\varepsilon$-FeOOH phase could undergo partial dehydrogenation and provide evidence that a new form of $\varepsilon$-FeOOHx ($0 < x < 1$) can be stabilized at lower mantle conditions. Similar to pyrite-type $FeO_2Hx$ ($0 < x < 1$), $\varepsilon$-FeOOHx may exist as a solid solution of $\varepsilon$-FeOOH and $FeO_2$ [8].

## 2. Materials and Methods

We started our experiment by synthesizing $\varepsilon$-FeOOH from goethite ($\alpha$-FeOOH, CAS: 20344-49-4). Goethite comprises double chains of edge-shared octahedra that form $2 \times 1$ channels hosting hydrogen bonds at an ambient condition. Above 5 GPa, it transforms to a high-pressure phase of $\varepsilon$-FeOOH, which consists of corner-shared single bands of octahedral. $\varepsilon$-FeOOH is thermodynamically stable in cold subducted slabs [25]. It is quenchable to ambient conditions once the crystal is synthesized in a multi-anvil press [26,27]. We followed the recipe by Suzuki [26,27] by compacting goethite powder in a gold capsule, rolling in a rhenium heater, and placing it in a Kawai-type multi-anvil press at the Geophysical Laboratory, Carnegie Institution of Washington. The use of the gold capsule was to seal water from out-of-capsule diffusion. The $\varepsilon$-FeOOH sample was synthesized at 10 GPa and 800 °C for 4 hours. After quenching to ambient conditions, the recovered samples were examined and confirmed on a diffractometer (Figure 1). The lattice parameters and volume at ambient conditions of the synthesized $\varepsilon$-FeOOH phase (orthorhombic, $P2_1nm$) were listed in Table 1 when compared with $\varepsilon$-FeOOH synthesized in other laboratories [26]. We found the ambient lattice parameters of our sample were consistent with other studies, which confirmed the composition of the synthesized $\varepsilon$-FeOOH was fully hydrogenated.

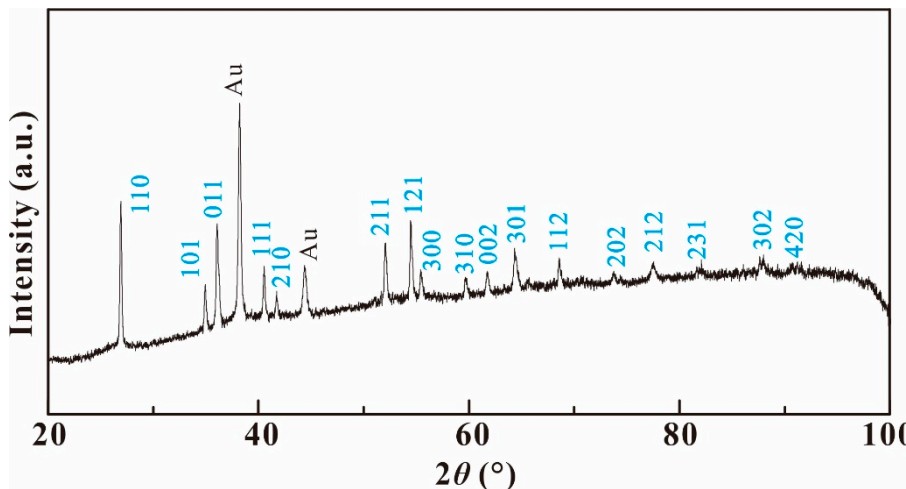

**Figure 1.** X-ray diffraction pattern of synthetic $\varepsilon$-FeOOH at ambient conditions. The sample was checked under a Brucker D8 diffractometer with a wavelength of 1.5406 Å. The diffraction pattern shows a pure phase of $\varepsilon$-FeOOH sample along with residual Au from the capsule.

**Table 1.** Structural data for the synthesized $\varepsilon$-FeOOH under an ambient environment.

| Phase | $V_0/Z$ (Å$^3$) | Lattice Parameters (Å, Degree) | | | Reference |
|---|---|---|---|---|---|
| | | *a* | *b* | *c* | |
| $\varepsilon$-FeOOH | 33.139(3) | 4.9544(2) | 4.4594(3) | 2.9999(1) | Suzuki et al. 2016 [26] |
| $\varepsilon$-FeOOH | 33.10(3) | 4.954(1) | 4.4540(9) | 3.0001(8) | Suzuki et al. 2009 [27] |
| $\varepsilon$-FeOOH | 32.636(3) | 5.273(4) | 4.423(3) | 2.798(2) | This study |

We then loaded powder $\varepsilon$-FeOOH in a diamond anvil cell (DAC) with Ne as the pressure medium. A second piece of $\varepsilon$-FeOOH from the same source was loaded without any pressure medium to check the effect of anisotropic compression [28]. The sample chamber was built by drilling a 100 to 120 μm hole in a tungsten gasket, which was squeezed between two diamond anvils with a 260 μm culet. Angular dispersive X-ray diffraction (XRD) patterns were obtained at the 13BM-C station of the GeoSoilEnviroCARS at the Advanced Phonon Source, Argonne National Laboratory (Argonne, IL, USA). The wavelength of the incident X-ray was 0.434 Å and the initial data reduction was performed by the Dioptas program [29]. We chose both ruby and gold for pressure calibration [30,31].

## 3. Results

In Figure 2, the high-pressure XRD patterns can be indexed to $\varepsilon$-FeOOH, gasket, and ruby. We collected XRD patterns up to 56.3 GPa and $\varepsilon$-FeOOH is still stable. Figure S1 (in Supplementary Materials) shows the selected XRD pattern of compared $\varepsilon$-FeOOH without a pressure medium. The molar volume of $\varepsilon$-FeOOH in Ne pressure medium is 24.7(3) Å$^3$ at 56.3 GPa, which is consistent with literature [32]. The volumes as a function of pressure were plotted in Figure 3a. Under a hydrostatic condition, the change of volume became much more incompressible at around 45 GPa, which may attribute to the spin transition of iron [32,33]. Likewise, spin transition occurred about 4–5 GPa in advance under non-hydrostatic conditions, which is possibly affected by the deviatoric stress [32]. In Figure 3b, we showed the pressure dependence of the relative lattice constants of $\varepsilon$-FeOOH. By comparing their evolution of the edge length along the lattice axis, we found the *a* and *b* axes were more affected by hydrostaticity and, thus, became more compressible under non-hydrostatic conditions. The elastic anisotropy may be the result of pressure induced hydrogen bond symmetrization in $\varepsilon$-FeOOH [32]. The centering of H atoms in two O atoms induced the shortening of O-O distance along the *b* axis. An interesting coincidence was the same H-bond behavior in the iso-structured $\delta$-AlOOH, whose thermal stability is significantly improved due to the H-bond symmetry [34]. The same symmetrical hydrogen bonds in $\varepsilon$-FeOOH may also expand its thermal stability field to higher temperatures.

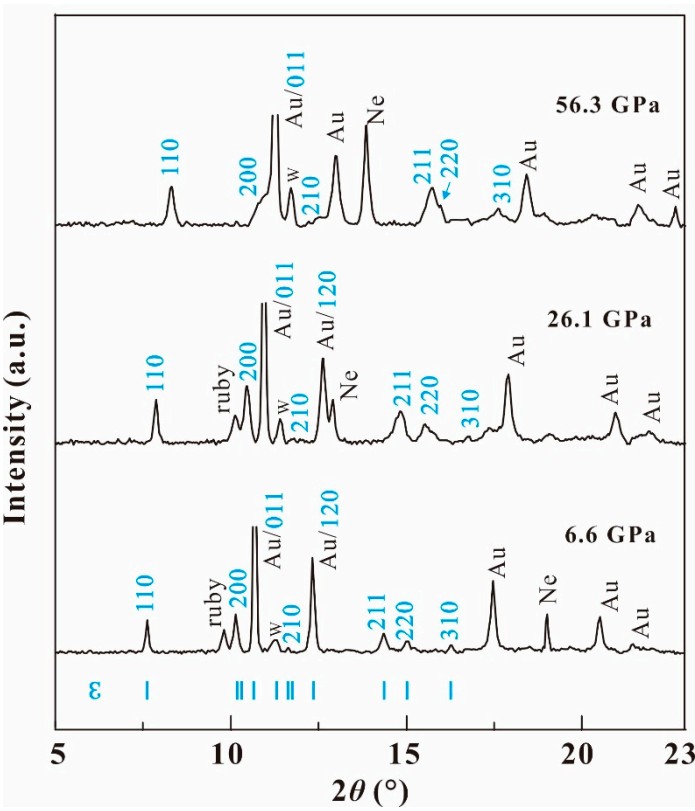

**Figure 2.** Selected x-ray diffraction patterns of compressed $\varepsilon$-FeOOH in Ne.

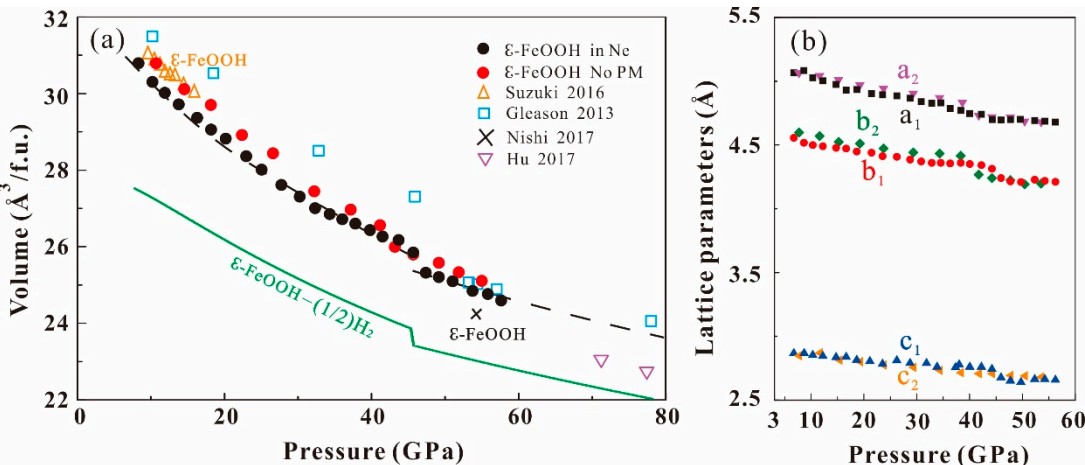

**Figure 3.** Equation of state for $\varepsilon$-FeOOH. (**a**) Volume versus pressure for $\varepsilon$-FeOOH in this study and literatures. Pressure-volume data for their high-spin or low-spin states were fitted to the third order Birch-Murnaghan equation of states. "X" and reversed triangle symbol were $\varepsilon$-FeOOH obtained by reheating Py-FeO$_2$H$x$ out of its stability range. Open spheres were fully hydrogenated $\varepsilon$-FeOOH. Pressure was calibrated by ruby and Au (gold) with up to ± 1 GPa uncertainty. (**b**) The lattice parameters of $\varepsilon$-FeOOH as a function of pressure under different pressure environments. $a_1$, $b_1$, and $c_1$ for hydrostatic conditions and $a_2$, $b_2$, and $c_2$ for non-hydrostatic conditions. PM, pressure medium.

The elastic parameters were derived by the fitting of *P*–*V* data to the third order Birch-Murnaghan equation of states (EOS).

$$P = \frac{3}{2}K_{T0}[(\frac{V_0}{V})^{\frac{7}{3}} - (\frac{V_0}{V})^{\frac{5}{3}}] \times \{1 + \frac{3}{4}(K_0' - 4)[(\frac{V_0}{V})^{\frac{2}{3}} - 1]\} \tag{1}$$



where $P$ is the pressure, $K_{T0}$ is the isothermal bulk modulus, $V_0$ and $V$ are the volumes at high pressure and ambient condition, respectively, and $K'_0$ is the pressure derivative of $K_{T0}$ at 1 bar. The data were listed in Table 2 [35]. It should be noted that the EOS of $\varepsilon$-FeOOH was separated to two regions due to the high-spin to low-spin transition. For the high-spin phase, the bulk modulus $K_0$ equals to 125.2(4) GPa with $V_0$ per formula unit (f.u.) of 32.4(4) Å at hydrostatic condition, which was consistent with previous studies [26]. The low-spin state has a much higher bulk modulus $K_0$ = 248.5(2) GPa. Consequently, the sample became more stiffened and incompressible in the low-spin state. The parallel non-hydrostatic experiments generally reproduced the results from hydrostatic conditions, with slightly lower bulk modulus of $K_0$ = 114.3(8) GPa in a high spin (Table 2). While the calculated $V_0$ is almost the same, a smaller $K_0$ from non-hydrostatic compression means that anisotropic compression may facilitate the formation of symmetric H bond in compressed $\varepsilon$-FeOOH.

**Table 2.** Compressibility parameters of $\varepsilon$-FeOOH. Abbreviations: PM, Pressure medium. AH, asymmetric O-H bonding. SH, symmetric O-H-O bonding. Calc., first-principles calculation.

| Method/PM | Note | $K_0$ (GPa) | $K_0'$ | $V_0/Z$ ($\text{Å}^3$) | Reference |
|-----------|------|-------------|--------|------------------------|-----------|
| Calc. | AH/high spin | 188(4) | 5.19(12) | 28.7(2) | Thompson et al. 2017 [35] |
| Calc. | SH/low spin | 223(2) | 4.07(3) | 29.3(1) | Thompson et al. 2017 [35] |
| No PM | high spin | 124(4) | 4 | 33 | Gleason et al. 2013 [32] |
| No PM | low spin | 162 | 4 | - | Gleason et al. 2013 [32] |
| NaCl | high spin | 126(3) | 10(1) | 33.1(3) | Suzuki et al. 2009 [27] |
| NaCl | high spin | 135(3) | 6.1(9) | 33.1(3) | Suzuki et al. 2016 [26] |
| Ne | high spin | 125.2(4) | 3.9(2) | 32.4(4) | This study |
| Ne | low spin | 248.5(2) | 3.9(8) | 29.1(2) | This study |
| no PM | high spin | 114.3(8) | 3.7(8) | 32.6(7) | This study |

The equation of the state is the key to study the diffusion and transportation behaviors of H in $\varepsilon$-FeOOH. Our previous simulation results suggested that $\varepsilon$-FeOOH with 0–75% of H defects is still energetically stable and the H loss may promote its phase transition to the pyrite-type FeOOH by lowering the transition barrier [36]. Removing all the hydrogen in $\varepsilon$-FeOOH leads to a $FeO_2$ stoichiometry, which is a pyrite-type phase synthesized above 74 GPa [8,37]. However, $FeO_2$ is also reported to have a few low-pressure polymorphs. For example, below ~50 GPa, the pyrite-$FeO_2$ may transform to an orthorhombic $FeO_2$ with *Pnnm* [38] or *Pbcn* [39] phase, which are both very similar to the crystal structure of $\varepsilon$-FeOOH. Would it be possible to form a complete solid-solution of $\varepsilon$-FeOOH and the corresponding orthorhombic $FeO_2$ [8]? While $\varepsilon$-FeOOH synthesized from $\alpha$-FeOOH and multi-anvil press is guaranteed to be fully hydrogenated, $\varepsilon$-FeOOH transformed from the pyrite-$FeO_2Hx$ might inherit the hydrogen loss from its high-pressure polymorph. We took data from our previous experiments [22] and compared it with the compression curve of $\varepsilon$-FeOOH. In Figure 4, we used the EOS of $\varepsilon$-FeOOH as the cap line of the fully hydrogenated phase. The volumes by subtracting 0.5 mole of hydrogen (Phase I [40]) from $\varepsilon$-FeOOH served as the baseline for hydrogen depleted $\varepsilon$-FeOOHx ($x = 0$). By comparing their molar volumes, we can calculate the H content $x$ for possible H depletion in $\varepsilon$-FeOOHx through the following equation.

$$x = [V_x - V_{e0}]/V_H \tag{2}$$

where $V_x$ is the observed volume of $\varepsilon$-FeOOHx from the experiment, $V_{e0}$ is the volume of hydrogen depleted $\varepsilon$-FeOOHx (the baseline with $x = 0$), and $V_H$ is the volume per formula unit of phase-I solid $H_2$, which is stable up to the pressures of Earth's core-mantle boundary [41]. The unit cell volumes of phase-I $H_2$ was obtained from Reference [40]. The same scheme was widely used in estimating hydrogen numbers in iron hydrides [24]. In Table 3, we compiled the volumes of FeOOHx and evaluated $x$ from our previous experiments and the literature [9,22,36]. The values of $x$ are within the range of 0.47–0.75, which are in the same range of our previous estimated $x$ in pyrite-type $FeO_2Hx$ [22].

The coincidence suggests the loss of H in the recrystallized $\varepsilon$-FeOOH$x$ samples may come from its high-pressure polymorph. We also noticed that the volume of $\varepsilon$-FeOOH reported by Nishi et al. [9] was 2.2% to 2.7% below the capline of $\varepsilon$-FeOOH, which is equivalent to $0.69 < x < 0.74$. It might be due to a dehydrogenation process by reheating a pyrite-type FeOOH sample at a relatively low pressure. We summarized the transportation of H in the polymorphic transition between $\varepsilon$-FeOOH$x$ and pyrite-type FeO$_2$H$x$ in Figure 4, where the O-H-O framework is re-established in $\varepsilon$-FeOOH$x$ with the same amount of H defects. As a result, $\varepsilon$-FeOOH$x$ can be regarded as a solid solution of $\varepsilon$-FeOOH and orthorhombic FeO$_2$. Even if there is no evidence that temperature-induced dehydrogenation will occur in $\varepsilon$-FeOOH, the H loss may be exhibited as an intrinsic property of pyrite-type FeO$_2$H$x$.

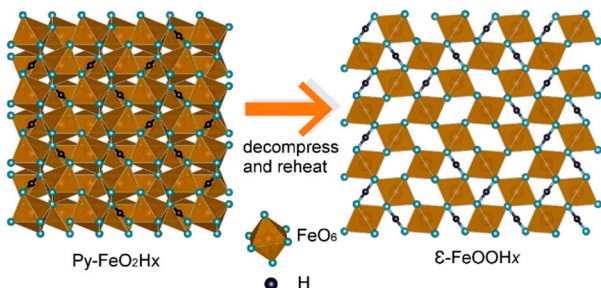

**Figure 4.** A schematic figure showing partially dehydrogenated $\varepsilon$-FeOOH$x$. The $\varepsilon$-FeOOH$x$ phase inherits the hydrogen loss from pyrite-type FeO$_2$H$x$.

**Table 3.** Hydrogen content $x$ in $\varepsilon$-FeOOH$x$.

| Pressure (GPa) | $V_0/f.u.$ (A$^3$) | $x$ | Reference |
|---|---|---|---|
| 72 | 23.04 | 0.47 | Hu et al. 2017 [22] |
| 78 | 22.76 | 0.46 | Hu et al. 2017 [22] |
| 87.5 | 23.15 | 0.75 | Zhu et al. 2017 [36] |
| 52 | ~24.2 | 0.69 | Nishi et al. 2017 [9] |

## 4. Discussion

During the transformation from $\varepsilon$-FeOOH to the pyrite structured FeO$_2$H, the emission of H is still a controversy in experiments [9,22]. The current work used fully hydrogenated $\varepsilon$-FeOOH as the cap line that was derived from the starting composition and compared with the volumes of $\varepsilon$-FeOOH$x$. In comparison, our previous work on pyrite-type FeO$_2$H$x$ used the baseline of FeO$_2$ [22]. The volume added to the baseline might be slightly underestimated due to the volume collapse amid the phase transition. Therefore, our current estimation on $\varepsilon$-FeOOH$x$ was more reliable and sensitive to H loss. The results confirmed both $\varepsilon$-FeOOH$x$ and pyrite-type FeOOH$x$ are partially dehydrogenated.

Our EOS of $\varepsilon$-FeOOH under different hydrostatic environments extended the studies of $\varepsilon$-FeOOH [26,27,32] to a wider pressure range. The XRD experiments of $\varepsilon$-FeOOH and $\varepsilon$-FeOOH$x$ have many implications for hydrogen diffusion and transportation in the deep Earth. H loss in transformed $\varepsilon$-FeOOH indicated H transportation is not affected by the phase transition. Therefore, recycled hydrogenated phases near the core-mantle boundary will retain the loss of hydrogen when they were subducted to a shallower part of the lower mantle even when they transformed to other phases like $\varepsilon$-FeOOH. It indicates $\varepsilon$-FeOOH in the lower mantle may have a significant amount of H defects. The total H content in the lower mantle may not be estimated by the H amount in full hydrogenated DHP. Instead, the case of dehydrogenation needs to be considered. Our experimental results support the H loss phenomenon in the high-pressure phases of FeOOH. The released hydrogen may escape or dissolve in the surrounding Earth materials as a part of the hydrogen cycle in Earth's deep interiors.

**Supplementary Materials:** The following are available online at http://www.mdpi.com/2073-4352/9/7/356/s1, Figure S1: Selected x-ray diffraction patterns of compressed $\varepsilon$-FeOOH under non-hydrostatic condition.

**Author Contributions:** Y.Z., Z.C., and D.Z. conducted the experiment. R.T. synthesized the sample. Y.Z., J.L., and Q.H. wrote the paper. Q.H. conceived the work. All authors discussed the results.

**Funding:** GeoSoilEnviroCARS (GSECARS) is supported by National Science Foundation - Earth Sciences (EAR-1128799) and Department of Energy - Geosciences (DE-FG02-94ER14466). YZ is supported by China Postdoctoral Science Foundation with grant 18NZ021-0213-216308. Operations of Center for High Pressure Science and Technology Advanced Research (HPSTAR) is partially supported by NSAF (Grant No: U1530402).

**Acknowledgments:** The authors acknowledge the use of synchrotron X-ray diffraction at the 13BM-C of GSECARS, Advanced Photon Source, Argonne National Laboratory.

**Conflicts of Interest:** The authors declare no conflict of interest.

**Angular Dispersive X-ray Diffraction Experiments:** Angular dispersive x-ray diffraction experiments were performed at 13BM-C station of the GSECARS at the APS, ANL. Samples of $\varepsilon$-FeOOH powders were grinded and pre-compressed to ~15 (T) × 60 (W) × 60 (L) $\mu m^3$ before placing on the culet of DAC. High pressure was achieved by using diamond anvils with 260 μm culet diameter. The sample chamber was a drilled hole with 100–120 μm diameter in a tungsten gasket. For quasi-hydrostatic condition, neon gas was pumped into the sample chamber using a gas-loading system at HPSTAR. For non-hydrostatic condition, no pressure medium was used. One or two pieces of ruby were placed around the sample to calibrate pressure. The ruby pressure scale was compared with the equation of state of gold [30] and Ne [42]. Pressure uncertainty is up to ± 1 GPa throughout the experiment.

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
