# Peer review of "Experimental Evidence for Partially Dehydrogenated ε-FeOOH"

_crystals, doi:10.3390/cryst9070356_

Round 1
Reviewer 1 Report
This is a very concise report on experimental results on partial dehydrogenation of ε-FeOOH. It is a very good contribution to our understanding of deep mantle processes. Although the authors' work is itself good, they may need some minor changes as follows.
For the convenience for wider scientific community, the process to obtain the Fe-O-H and Al-O-H system in the deep mantle should be referred to by using a few sentences in Introduction.This may be obvious for the community of UHP materials but not so familiar for the people working on lower-pressure earth materials.
I recommend the authors to change the closing remark (lines 183-174). They are required to state more focused prospective based on their experimental result.
Reviewer 2 Report
The submitted manuscript presents experimental evidence for partially dehydrogenated
ε-FeOOH mineral. For this purpose, angular dispersive x-ray diffraction was used.
The authors found that ε-FeOOH transformed from pyrite-FeO2Hx may inherit the hydrogen loss that occurred at the pyrite-phase.
Presented analyses were performed with appropriate standards. Materials and methods were described well. Discussion is interesting and could be important for research in this area. High pressure studies of hydrogen diffusion and transportation are important to understand the crystallographic properties of hydrous minerals.
The Birch-Murnaghan equation of states was employed for the study, and the obtained parameters were discussed. This equation should be given in the manuscript.
Reviewer 3 Report
The authors " ... calculate the H content x for possible H136 depletion in ε-FeOOHx through the following equation:137 x = [Vx Ve0]/VH (1)
where Vx is the observed volume of ε-FeOOHx from experiment, Ve0 is the volume of hydrogen139 depleted ε-FeOOHx (the baseline with x=0), VH is the volume of 0.5H2" . However, hydrogen is quite small to induce significant and accurately measurable volume changes. Moreover, it is crude to adopt a hard-sphere description such as: "VH is the volume of 0.5H2"; due to its small mass, its wavefunction intrudes the surrounding potential walls - so, what is defined as VH ?!
Alternatively, the authors could have clear signaature of hydrogen by performing IR spectroscopy in their DAC. Additionally, direct electrical conductivity cound have been made in the same apparatus.
Round 2
Reviewer 3 Report
The revised manuscript is acceptable for publication.